# S7: Selective and Simplified State Space Layers for Sequence Modeling

## Abstract

A central challenge in sequence modeling is efficiently handling tasks with extended contexts. While recent state-space models (SSMs) have made significant progress in this area, they often lack input-dependent filtering or require substantial increases in model complexity to handle input variability. We address this gap by introducing **S7**, a simplified yet powerful SSM that can handle input dependence while incorporating stable reparameterization and specific design choices to dynamically adjust state transitions based on input content, maintaining efficiency and performance. We prove that this reparameterization ensures stability in long-sequence modeling by keeping state transitions well-behaved over time. Additionally, it controls the gradient norm, enabling efficient training and preventing issues like exploding or vanishing gradients. S7 significantly outperforms baselines across various sequence modeling tasks, including neuromorphic event-based datasets, Long Range Arena benchmarks, and various physical and biological time series. Overall, S7 offers a more straightforward approach to sequence modeling without relying on complex, domain-specific inductive biases, achieving significant improvements across key benchmarks.

## 1 Introduction

Sequence modeling is a fundamental challenge in deep learning, with applications spanning natural language processing, computer vision, audio processing, and genomics (Sutskever et al., 2014; Graves et al., 2013). The core problem lies in effectively capturing and utilizing information from long input sequences while maintaining computational efficiency. Traditional approaches, such as recurrent neural networks (RNNs) (Hochreiter & Schmidhuber, 1997), struggle with long-range dependencies due to vanishing gradients (Bengio et al., 1994), while attention-based models like Transformers (Vaswani et al., 2017) face quadratic complexity in sequence length, limiting their scalability. While efficient, convolutional models (Bai et al., 2018) cannot often capture global context. The key challenge is to design a model that can (1) efficiently process very long sequences, (2) adaptively filter and retain relevant information over extended time horizons, (3) perform content-based reasoning, and (4) maintain a compact state representation. Recent advances in Deep State Space Models (Deep SSMs) (Gu et al., 2020; Hasani et al., 2020) have shown promise, but existing approaches like S4 (Gu et al., 2022a) and Mamba (Gu & Dao, 2023) still face limitations in balancing these requirements. S4 models, while efficient, lack input-dependent filtering capabilities, and Mamba, though more flexible, introduces significant complexity. There is a clear need for a model that combines the efficiency of recurrent architectures with the adaptive, content-aware processing capabilities of more complex models without sacrificing simplicity or generalizability across diverse sequence modeling tasks (Tay et al., 2022; Schlag et al., 2021).

The importance of effective sequence modeling cannot be overstated in today's AI landscape. It forms the backbone of large language models (Brown et al., 2020), which have revolutionized natural language processing and are increasingly applied across diverse domains. In computer vision, sequence modeling enables video data processing and event-based vision (Zubić et al., 2024a; 2023), critical for applications like autonomous driving and robotics. Genomics allows for analyzing long DNA sequences, potentially unlocking breakthroughs in personalized medicine and drug discovery (Avsec et al., 2021). However, the problem is inherently challenging due to several factors. First, the sheer length of sequences in real-world applications (often millions of tokens) makes it computationally intensive to process and retain relevant information (Tay et al., 2021a). Second, the relevance

of information can vary dramatically across the sequence, requiring adaptive filtering mechanisms (Katharopoulos et al., 2020). Third, capturing long-range dependencies and performing content-based reasoning demands sophisticated architectures to maintain and update a meaningful state over time (Dai et al., 2019). Finally, there's a fundamental tension between model expressivity and computational efficiency – more powerful models often come at the cost of increased complexity and resource requirements (Tay et al., 2021b). Balancing these competing demands while maintaining generalizability across diverse tasks remains an open challenge in the field (Schlag et al., 2021), driving the need for innovative approaches to push the boundaries of what's possible in sequence modeling.

Recent advancements in sequence modeling have made significant strides. Transformer architectures (Vaswani et al., 2017) revolutionized the field with their attention mechanisms, enabling parallel processing and capturing long-range dependencies. However, their quadratic complexity in sequence length remains a limitation. Efficient transformer variants (Kitaev et al., 2020; Beltagy et al., 2020) attempted to address this, but often at the cost of reduced model capacity. The emergence of Deep State Space Models (SSMs) marked a new frontier, with S4 (Gu et al., 2022a) demonstrating impressive performance on long-range tasks while maintaining linear complexity. Mamba (Gu & Dao, 2023) further improved upon this by introducing input-dependent dynamics, enhancing the model's ability to perform content-based filtering. Despite these advances, the field has yet to achieve an optimal balance between efficiency, adaptability, and simplicity. The primary stumbling block lies in reconciling the need for input-dependent processing—crucial for adaptive filtering and content-based reasoning—with the computational efficiency of recurrent architectures. S4 models, while efficient, lack input-dependent dynamics, limiting their ability to adapt to varying content. Conversely, Mamba introduces input dependence at the cost of increased complexity and reliance on specialized hardware implementations. The challenge now is to develop a model that combines the strengths of these approaches—the efficiency and simplicity of recurrent models with the adaptive capabilities of input-dependent systems—without compromising on performance or generalizability across diverse tasks (Schlag et al., 2021; Tay et al., 2022; Zubić et al., 2024b). This balance is critical for pushing sequence modeling towards more general and scalable AI systems capable of handling the complexities of real-world data across various domains.

Our paper introduces S7, a simplified yet powerful State Space Model (SSM) that advances the frontier of sequence modeling by making the purely recurrent, time-domain S5 model input-dependent. This critical insight combines the efficiency of recurrent architectures with the adaptive processing capabilities of more complex models. By dynamically adjusting state transitions based on input content, S7 performs content-based reasoning and adaptive filtering while preserving recurrent models' simplicity and computational efficiency. Unlike S4, which lacks input-dependent dynamics, or S6 (Mamba), which introduces hardware-specific complexity, S7 achieves a balanced design. We introduce stable reparameterization and additional design choices that ensure long-term stability and performance across diverse tasks.

Our extensive experiments validate S7's versatility and effectiveness across a wide range of sequence modeling tasks, setting new standards in the field. On event-based vision datasets, S7 achieves state-of-the-art results, attaining accuracies of 99.2% on DVS-Gesture, 96.3% on Spiking Heidelberg Digits, and 88.2% on Spiking Speech Commands, significantly outperforming traditional dense methods. In human activity recognition, S7 achieves an impressive accuracy of 94.1%, demonstrating its capability to handle irregularly sampled, noisy time-series data. For genomics classification, S7 sets a new benchmark with 97.5% accuracy on the EigenWorms dataset, effectively capturing very long-term dependencies in sequences of length 17,984. On the Long Range Arena benchmarks (Tay et al., 2021a), S7 excels in multiple tasks, achieving 63.77% accuracy on ListOps and 91.80% on Retrieval, outperforming prior state-of-the-art models and 87.22% accuracy on the Text classification task, showcasing its ability to process and understand long textual sequences. Moreover, S7 demonstrates remarkable efficiency and precision in simulating physical dynamical systems, reducing the test $L^2$ error by nearly half compared to previous models in predicting the FitzHugh-Nagumo system, and achieves the lowest Mean Squared Error (MSE) of 0.114 in the Walker2d Kinematic Simulation task. These results show S7's ability to generalize across diverse domains, offering a more efficient and adaptable approach to sequence modeling without relying on domain-specific inductive biases, and highlight S7's improvements in capturing long-range dependencies and complex temporal patterns while maintaining computational efficiency, marking a significant improvement

over previous models and opening new avenues for research and application in the field of sequence modeling.

## 2 RELATED WORK

Sequence modeling has evolved from traditional RNNs (Elman, 1990), including LSTMs (Hochreiter & Schmidhuber, 1997) and GRUs (Cho et al., 2014), which struggle with long-range dependencies (Bengio et al., 1994), to CNNs adapted for sequential data (Bai et al., 2018; van den Oord et al., 2016), and then to attention-based models like Transformers (Vaswani et al., 2017). While Transformers excel at capturing long-range dependencies, their quadratic complexity led to the development of efficient variants using linear (Katharopoulos et al., 2020; Wang et al., 2020) or sparse attention (Child et al., 2019; Beltagy et al., 2020). SSMs emerged as a promising approach, with S4 (Gu et al., 2022a) achieving state-of-the-art performance on long-range tasks while maintaining linear complexity. Subsequent work refined SSMs, leading to S4D (Gu et al., 2022b) and S5 (Smith et al., 2023). The limitation of fixed dynamics in traditional SSMs motivated input-dependent models, notably the Mamba architecture (Gu & Dao, 2023) with its selective state spaces and its recent extension, Mamba-2 (Dao & Gu, 2024), which further improves performance and efficiency. These advancements have impacted various domains, including event-based vision processing (Zubić et al., 2024a) and have been evaluated on long-range sequence modeling benchmarks (Tay et al., 2021a). Theoretical work has explored connections to control theory (Gu et al., 2021), approximation capabilities (Gu et al., 2020), and complexity analysis (Dao et al., 2022). Our work, S7, builds upon these foundations, particularly SSMs and input-dependent models, aiming to combine the efficiency of recurrent architectures with adaptive capabilities to address limitations in existing approaches. Specifically, S7 applies to S5 the same principle of input-dependence that Mamba introduced to S4, but within the context of a purely recurrent, time-domain model.

## 3 METHOD

### 3.1 BACKGROUND

**State Space Models (SSMs)** SSMs are a class of models widely used in control theory, neuroscience, and machine learning for modeling sequential data. The core of SSMs lies in their representation of a system's evolution over time through a latent state. Mathematically, SSMs are typically represented as:

$$\dot{x}(t) = Ax(t) + Bu(t) \qquad y(t) = Cx(t) + Du(t) \qquad (1)$$

where $x(t) \in \mathbb{R}^H$ is the latent state vector, $u(t) \in \mathbb{R}^N$ is the input signal, and $y(t) \in \mathbb{R}^N$ is the output. The system is governed by the matrices $A \in \mathbb{R}^{H \times H}$, $B \in \mathbb{R}^{H \times N}$, $C \in \mathbb{R}^{N \times H}$, and $D \in \mathbb{R}^{N \times N}$, which are the parameters to be learned. SSMs capture long-range dependencies in sequential data by evolving the latent state over time in a continuous manner (Gu et al., 2020; Smith et al., 2023). In deep learning, SSMs can be stacked in multiple layers, allowing them to process complex sequential data more effectively. By stacking SSM layers, these models can capture intricate temporal patterns while maintaining a compact state representation, efficiently handling long sequences (Gu et al., 2022a).

**Discretization of Continuous SSMs** In practice, continuous SSMs must be discretized to apply them in computational models, particularly for deep learning tasks. The discretization process converts continuous-time dynamics into a form that can be computed at discrete time steps, typically using methods such as the zero-order hold (ZOH). The discrete equivalent of the continuous system is given by:

$$x_k = \bar{\Lambda}x_{k-1} + \bar{B}u_k \qquad y_k = \bar{C}x_k + \bar{D}u_k \qquad (2)$$

where $\bar{\Lambda} = e^{A\Delta t}$ and $\Delta t$ is the time step size (Smith et al., 2023). This formulation allows the model to process input sequences at discrete intervals, making it suitable for training on modern hardware. Efficient discretization techniques are essential to ensure that SSMs retain their ability to model long-range dependencies without becoming computationally expensive (Gu et al., 2022a).

## 3.2 INPUT DEPENDENCY IN STATE-SPACE MODELS

To improve the performance of SSMs, input dependence can be introduced by making the transition matrices a function of the input. In S7, the system evolution at time step $k$ can be described by the following discretized equations:

$$x_k = \bar{\Lambda}_k x_{k-1} + \bar{B}_k u_k \qquad y_k = \bar{C}_k x_k + \bar{D}_k u_k \tag{3}$$

Here, the transition matrix $\bar{\Lambda}_k$, along with the input matrices $\bar{B}_k$, $\bar{C}_k$, and $\bar{D}_k$, are functions of the input $u_k$, allowing the model to adapt to the current input at each time step dynamically. This enables the model to filter information, selectively determining what to retain and forget. Doing so enhances the model's ability to capture essential long-term dependencies while filtering out irrelevant information, improving performance and generalization. The system output $y_k$ is processed through normalization layers, followed by a GeLU activation and a gating mechanism. The gating function, represented by a sigmoid activation $\sigma(W \cdot \text{GeLU}(y_k))$, helps regulate how much of the processed information passes through, enabling the model to control the flow of information based on the input signal and current state.

This dynamic gating allows the model to adjust the information flow based on the input signal and the current state, providing a more robust and flexible state evolution. Introducing input-dependent dynamics improves S7's ability to handle diverse temporal dependencies, effectively filtering and retaining relevant information over time. By making the state transition matrices depend on the input $u_k$, S7 improves on the limitations of static state transitions found in previous models, such as S4 and S5 (Gu et al., 2022a; Smith et al., 2023), which lacked the flexibility to adapt state transitions based on the input. This selective updating of internal states allows S7 to balance long-term and short-term dependencies, leading to better performance and more effective memory management in sequence modeling tasks.

## 3.3 THE S7 LAYER

Building on the foundation of input dependency and recurrent SSMs, we introduce the **S7** model, which extends the capabilities of the S5 model by incorporating input-dependent state transitions and improving training stability via reparameterization techniques. This allows S7 to dynamically adjust its state transitions based on input content while maintaining the efficiency of recurrent models for long-sequence tasks.

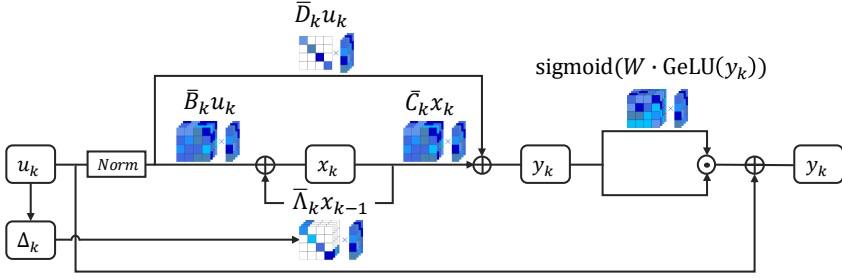

Figure 1: **The S7 Layer Architecture.** The diagram illustrates the recurrent structure of the S7 model, which integrates input-dependent state-space models with stable parameterization. The transition matrices $B_k$, $C_k$, $D_k$, and $\bar{\Lambda}_k$ reflect the interaction between input $u_k$ and previous hidden state $x_{k-1}$, while non-linearity is reinforced by the sigmoid. Contrary to input-dependent S6 (Mamba) Gu & Dao (2023), this model is much simpler and based on S5 (Smith et al., 2023).

**Stable Reparameterization for Long-Term Dependencies** To ensure stability during long-sequence modeling, with **S7**, we employ a reparameterization of the transition matrix $\bar{\Lambda}_k$, inspired by StableSSM (Wang & Li, 2024). The recurrent matrix is modified by a stability function, ensuring that the system avoids unstable behavior over time. Specifically, the reparameterization is applied as:

$$\bar{\Lambda}_k = f(\Lambda_k) = I - \left(\Lambda_k^2 + 0.5I\right)^{-1} \tag{4}$$

where $I$ is the identity matrix. This stability function guarantees that the eigenvalues of the matrix remain within a range that promotes stable dynamics, even in the presence of long-range dependencies. Theoretical and experimental details on reparametrizations are in the A.4 & A.6. As already said, we assume the system follows input-dependent dynamics, where the hidden states evolve according to the differential equation:

$$x_k = \Lambda_k(u_k; \theta_m)x_{k-1} + B_k(\theta_m)u_k + b_k(\theta_m) \quad y_k = C_k(\theta_m)x_k + D_k(\theta_m)u_k \quad (5)$$

where $x_k \in \mathbb{R}^m$ is the hidden state at time step $k$, and $u_k \in \mathbb{R}^d$ is the input at time step $k$. The matrices $\Lambda_k(\theta_m) \in \mathbb{R}^{m \times m}$, $B_k(\theta_m) \in \mathbb{R}^{m \times d}$, $b_k(\theta_m) \in \mathbb{R}^m$, $C_k(\theta_m) \in \mathbb{R}^{d \times m}$, and $D_k(\theta_m) \in \mathbb{R}^{d \times d}$ are parameterized by $\theta_m$, the model's trainable parameters.

The parameterization of the system $\theta_m$ in terms of our notation can be described as $\theta_m = (\Lambda_k, B_k, b_k, C_k, D_k)$, where $\theta_m \in \Theta_m := \{\mathbb{R}^{m \times m} \times \mathbb{R}^{m \times d} \times \mathbb{R}^m \times \mathbb{R}^{d \times m} \times \mathbb{R}^{d \times d}\}$. This defines $\theta_m$ as the set of all trainable parameters in the SSM.

**Assumption 3.1.** The mappings $\theta_m \mapsto \Lambda_k(\theta_m), \theta_m \mapsto B_k(\theta_m), \theta_m \mapsto b_k(\theta_m)$, and $\theta_m \mapsto C_k(\theta_m)$ are Lipschitz continuous for all $u_k$ in a bounded input space $\mathcal{X} \subset \mathbb{R}^d$. This ensures that small parameter changes lead to small changes in the state transition matrices, promoting stable learning and smooth transitions over time.

**Assumption 3.2.** For all $u_k \in \mathcal{X}$, the eigenvalues of $\Lambda_k(\theta_m)$ have negative real parts, ensuring that the system remains uniformly asymptotically stable.

**Assumption 3.3.** The parameters $\theta_m$ are subject to a stable reparameterization $f$, such that $\theta_m = f(w_m)$, meaning raw model $w_m$ parameters after reparametrization are $\theta_m$, and $f$ satisfies the stable reparameterization condition defined by:

$$\sup_w \left[ \|f(w)\| \sup_{\|\tilde{w}-w\| \leq \beta} \int_0^\infty \|\Phi_{\tilde{w}}(k, s) - \Phi_w(k, s)\| \, dk \right] \leq g(\beta) \quad (6)$$

for some continuous function $g : [0, \infty) \to [0, \infty]$ with $g(0) = 0$. Here, $\Phi_w(k, s)$ denotes the state transition matrix corresponding to parameters $w$, which satisfies:

$$\frac{d}{dk}\Phi_w(k, s) = \Lambda_k(u_k; f(w))\Phi_w(k, s), \quad \Phi_w(s, s) = I_m. \quad (7)$$

The state transition matrix $\Phi_w(k, s)$ describes the evolution of the system's state from time $s$ to time $k$ under the dynamics defined by $\Lambda_k(u_k; f(w))$ and $I_m$ is the identity matrix. The constant parameter $\beta$ limits how much the parameters can be perturbed while ensuring that the state transition matrices and system behavior remain stable. The function $g(\beta)$ helps quantify how much the difference between the perturbed and unperturbed system can grow. As $\beta \to 0$, this difference should vanish.

**Assumption 3.4.** The system's inputs $u_k$ and hidden states $x_k$ are uniformly bounded, and the matrices $\Lambda_k(\theta_m)$, $B_k(\theta_m)$, $b_k(\theta_m)$, and $C_k(\theta_m)$ are uniformly bounded in $m$.

**Theorem 3.5** (Existence of Stable Approximation by Stable Reparameterization with Input-Dependent Dynamics). *Let $\mathbf{H}$ be any bounded, causal, continuous, and regular linear functional. Suppose $\mathbf{H}$ is approximated by a sequence of state-space models $\{\widehat{\mathbf{H}}(\cdot; \theta_m)\}_{m=1}^\infty$ with input-dependent dynamics of the form Eq. 5. Then, the approximation of $\mathbf{H}$ by the sequence $\{\widehat{\mathbf{H}}(\cdot; \theta_m)\}_{m=1}^\infty$ is a stable approximation in the Sobolev-type norm defined by:*

$$\left\|\mathbf{H} - \widehat{\mathbf{H}}\right\|_{W^{1,\infty}} = \sup_k \left( \|H_k - \widehat{H}_k\|_\infty + \left\|\frac{dH_k}{dk} - \frac{d\widehat{H}_k}{dk}\right\|_\infty \right). \quad (8)$$

*Proof.* Here, we provide a brief sketch of the proof, with full details in the A.2.

The mappings from parameters $\theta_m$ to the system matrices $\Lambda_k(u_k; \theta_m)$, $B(\theta_m)$, $b(\theta_m)$, and $c(\theta_m)$ (with $c \in \mathbb{R}^m$ being small, assuming a single-output dimension for simplicity) are Lipschitz continuous, ensuring that small perturbations in $\theta_m$ lead to small changes in the system dynamics. The eigenvalues of $\Lambda_k(u_k; \theta_m)$ have negative real parts for all $u_k \in \mathcal{X}$, which guarantees uniform asymptotic stability of the system. As a result, the state transition matrix $\Phi(k, s; u, \theta_m)$ decays exponentially as $k - s$ increases, preserving stability over time. The stable reparameterization function $f$ further ensures that parameter perturbations are well-controlled, and the condition involving $g(\beta)$

implies that as $\beta \to 0$, the difference between the perturbed and unperturbed state transition matrices vanishes.

To analyze the approximation error, we bound the total error $E(\beta)$ by combining the error due to model capacity (which vanishes as $m \to \infty$) and the error from parameter perturbations. Applying Grönwall's inequality and using the Lipschitz properties of the mappings, we show that the error from perturbations is proportional to $\beta$. As $m \to \infty$ and $\beta \to 0$, the total approximation error tends to zero, ensuring that the sequence $\{\widehat{\mathbf{H}}(\cdot; \theta_m)\}_{m=1}^{\infty}$ provides a stable approximation of $\mathbf{H}$. □

**Theorem 3.6** (Parameterizations Influence the Gradient Norm Scale in Input-Dependent SSMs)**.**
*The gradient of the loss with respect to the trainable parameter $w_j$ satisfies the following bound:*

$$\left| \frac{\partial Loss}{\partial w_j} \right| \leq C_{\mathbf{H}, \widehat{\mathbf{H}}_m} \left| f'(w_j) \right|, \tag{9}$$

*where $f'(w_j)$ is the derivative of the reparameterization function $f$ with respect to $w_j$ and $C_{\mathbf{H}, \widehat{\mathbf{H}}_m}$ is a constant independent of $w_j$, but dependent on the target functional $\mathbf{H}$ and the model $\widehat{\mathbf{H}}_m$.*

*Proof.* We provide a brief proof sketch; detailed steps are in the A.3.

The goal is to bound the gradient of the loss function, which measures the difference between the target functional and the model's output. Since the target does not depend on the model parameters $w_j$, the model output determines the gradient entirely. This output depends on the parameterized functions $c(\theta_m)$, which are Lipschitz continuous, and the hidden state dynamics, which are stable under the given assumptions.

Using the Lipschitz continuity of $c(\theta_m)$ and the uniform stability of the system, we show that the gradient of the model output with respect to $w_j$ is bounded by a constant times the derivative of the reparameterization function $f$. This leads to the conclusion that the gradient of the loss function scales proportionally to $|f'(w_j)|$, explaining the role of the reparameterization in controlling optimization behavior. □

### 3.4 ADDITIONAL DESIGN CHOICES FOR EVENT-BASED NEUROMORPHIC TASKS

**Efficient Tokenization for Event-Based Data**    In S7, we introduce an event-based tokenization scheme that captures the neuromorphic data's spatial and temporal nature. This method utilizes a sensor of size $(s_x, s_y)$, where $s_x$ is the number of horizontal pixels and $s_y$ is the number of vertical pixels. Each event, $\varepsilon$, is defined by the following quadruple: $(x, y, t, p)$, where $x$ and $y$ represent the spatial coordinates of the event on the sensor, $t$ represents the timestamp of the event, and $p \in \{-1, 1\}$ is the polarity, indicating the nature of the event (positive or negative change). We then define a unique token for each event $\varepsilon$ using the following formula:

$$\mathcal{T}_{\text{S7}}(\varepsilon) = 2 \cdot (x \cdot s_x + y) + p \tag{10}$$

In this formula, $\mathcal{T}_{\text{S7}}(\varepsilon)$ denotes the token generated for the event $\varepsilon$ using the S7-specific tokenization scheme, as indicated by the subscript. This bijective mapping ensures each event produces a unique token, preventing collisions where different events could share the same token, as seen in models like EventSSM (Schöne et al., 2024). By encoding spatial and polarity information, the S7 scheme enhances the model's ability to efficiently process asynchronous, real-time data.

**Efficiency Through Event Pooling and Asynchronous Discretization**    Also, we optimize computational efficiency through *Event Pooling*, which pools hidden states over a window of size $p$, reducing computational load:

$$x_{k+p} = \Lambda_k^p x_k + \sum_{i=1}^{p} \Lambda_k^{p-i} B u_{k+i} \tag{11}$$

Further, *Asynchronous Discretization* updates the hidden state based on varying time intervals between events, enabling S7 to handle real-time event streams efficiently:

$$x_k = e^{\Lambda_k \Delta t_k} x_{k-1} + B u_k \tag{12}$$

This ensures that S7 remains efficient in processing asynchronous data, such as in neuromorphic vision and spiking neural networks. By integrating input dependence, stable reparameterization, and efficient tokenization, S7 enables significant performance improvement, surpassing its predecessors in performance and scalability.

# 4 EXPERIMENTS

We evaluate the performance of the proposed S7 model across several tasks. In Sec. 4.1, we describe the experimental setup, including training protocols and evaluation metrics, and in Sec. 4.2, we assess the model on neuromorphic event data. Sec. 4.3 focuses on long-range sequence modeling with the LRA benchmark (Tay et al., 2021a). Dynamical system prediction tasks, including Pendulum Regression, are explored in Sec. 4.4. Finally, in Sec. 4.5, we evaluate S7 on human activity recognition and genomics classification. In Sec. 4.6, we also explore the Walker2D kinematic simulation. In the A.6, we perform an ablation study to evaluate the importance of the reparameterization method in improving model performance.

## 4.1 EXPERIMENTAL SETUP

We follow the experimental setups for training and evaluation described in EventSSM (Schöne et al., 2024) for Sec. 4.2, S5 (Smith et al., 2023) for Sec 4.3, S5 & LEM (Rusch et al., 2022) for Sec. 4.4 and ODE-LSTM (Lechner & Hasani, 2020) & LEM for Sec. 4.5. Specifically, we use a cosine learning rate schedule for all datasets throughout the training process. Separate weight decay is applied to the SSM parameters to control regularization. We select the best validation epoch for final testing to ensure optimal performance. Cross-entropy is employed as the loss function for all tasks except Pendulum & FitzHugh-Nagumo system (Sec. 4.4) and Walker2D, for which we used MSE. All models are trained using Tesla V100 and Quadro RTX 8000 GPUs. The training code is implemented in JAX (Bradbury et al., 2018), while Tonic (Lenz et al., 2021) is used for fast event-based data loading. Additionally, we introduce a separate weight decay for the dense layers responsible for input dependence, allowing these layers to be fine-tuned independently of the core SSM parameters. This enables better control over regularization in the filtering layers and further leads to improved generalization and performance across different tasks.

## 4.2 EVENT (NEUROMORPHIC) DATASETS

We process raw, asynchronous event streams in these datasets, fully leveraging S7's ability to model long-range temporal dependencies directly from raw events. Unlike the majority of approaches in this context, which convert events into frames or other representations, only EventSSM (Schöne et al., 2024) and our S7 operate directly on the raw event data. This allows us better to capture the fine-grained dynamics unique to event-based data streams.

| Dataset | LSN | SGN | CNN+S5 | BET | EventSSM | S7 (Ours) |
|---|---|---|---|---|---|---|
| **DVS-Gesture** | - | - | 97.8 (6.8M) | 98.8 (-) | 97.7 (5.4M) | **99.2 (4.1M)** |
| **Spiking Heidelberg Digits** | 95.1 (0.2M) | 94.6 (3.9M) | 93.8 (3.9M) | - | 95.5 (0.4M) | **96.3 (0.5M)** |
| **Spiking Speech Commands** | 80.7 (2.5M) | 77.4 (3.9M) | 81.2 (4.2M) | - | 87.1 (0.6M) | **88.2 (0.6M)** |

Table 1: Accuracy comparison of LSN (Hammouamri et al., 2024), SGN (Bittar & Garner, 2022), CNN+S5, BET (Liu et al., 2022), EventSSM (Schöne et al., 2024), and S7 on event datasets. The number of model parameters (in millions) is shown in parentheses.

**DVS-Gesture** The DVS-Gesture dataset (Amir et al., 2017) features 11 hand gestures recorded by a DVS128 sensor at 128x128 resolution, with over 1,100 training samples and up to 1.5 million events per sequence. Following EventSSM's data augmentations (Schöne et al., 2024), we apply spatial-jitter, time-jitter, and CutMix. S7 achieves 99.2% accuracy, surpassing EventSSM (97.7%) and the best dense method BET (Liu et al., 2022) (98.8%). Full results are in Table 1.

**Spiking Heidelberg Digits (SHD)** The SHD dataset (Cramer et al., 2019) challenges models with 20 classes of spoken digits converted into spike trains. It includes 8,200 training samples with se-

quences having a median of 8,000 events. This dataset tests a model's ability to process event-based audio data. S7 achieved an accuracy of 96.3%, outperforming both the best dense method (95.1%) and EventSSM (95.5%), with only a slight increase in parameters (0.5M vs. 0.4M). Additionally, compared to dense methods such as LSN (Hammouamri et al., 2024) and SGN (Bittar & Garner, 2022), S7 demonstrates superior performance with far fewer parameters.

**Spiking Speech Commands (SSC)**    The SSC dataset (Cramer et al., 2019) includes 35 classes of spoken commands converted into spike trains, with sequences having a median of 8,100 events and a total of 75,500 training samples. We applied time-jitter, channel-jitter, and CutMix augmentations (Yun et al., 2019) for this large-scale dataset. S7 achieved 88.2% accuracy, outperforming both the best dense method (80.7%) and EventSSM (87.1%) while maintaining the exact parameter count (0.6M) as EventSSM and using up to 7x fewer parameters than the best dense models.

## 4.3 LONG RANGE ARENA

We adopt the setup used in the S5 framework (Smith et al., 2023), converting integer-tokenized datasets into event-based formats with regular time gaps, treating tokens as events with a polarity of 1. Experiments were conducted on all six Long Range Arena (LRA) tasks: ListOps, Text, Retrieval, Image, Pathfinder, and Path-X, with results summarized in Table 2. Among the models compared, S6 (Mamba) and our S7 are the only ones employing input-dependent dynamics. Our results demonstrate that S7 outperforms Mamba across the LRA benchmarks (71.82 vs 66.59 average), highlighting the effectiveness of our approach and establishing S7 as the best input-dependent model for long, challenging sequence modeling. Notably, S7 achieves state-of-the-art performance on the ListOps and Retrieval tasks, with accuracies of 63.77% and 91.80%, respectively. However, it is challenging for input-dependent models to surpass Linear Time-Invariant (LTI) methods like S5 on certain tasks because input-dependency can lead to forgetting some tokens, which hurts performance when precise retention of input information is crucial. While Mega achieves the highest average score overall, it operates with quadratic complexity in sequence length, making it less scalable for long sequences. In contrast, S7 offers a favorable trade-off between performance and scalability, achieving competitive results with linear computational complexity.

| Dataset | Mega | S4 | S5 | S6 (Mamba) | LRU | S7 (Ours) |
|---|---|---|---|---|---|---|
| ListOps | 63.14 | 59.60 | 62.15 | 38.02 | 60.20 | **63.77** |
| Text | **90.43** | 86.82 | 89.31 | 82.98 | 89.40 | 87.22 |
| Retrieval | 91.25 | 90.90 | 91.40 | 72.14 | 89.90 | **91.80** |
| Image | **90.44** | 88.65 | 88.00 | 69.82 | 89.00 | 61.14 |
| Pathfinder | **96.01** | 94.20 | 95.33 | 69.26 | 95.10 | 65.52 |
| Path-X | 97.98 | 96.35 | **98.58** | 67.32 | 94.20 | 61.50 |
| **Average** | **88.21** | 86.09 | 87.46 | 66.59 | 86.30 | 71.82 |

Table 2: Accuracy comparison of Mega (Ma et al., 2023), S4 (Gu et al., 2022a), S5 (Smith et al., 2023), S6 (Mamba) (Gu & Dao, 2023), LRU (Orvieto et al., 2023), and S7 across LRA tasks. The best result for each task is highlighted in **bold**, while the second-best result is underlined. The overall best and second-best results are similarly bolded and underlined in the average row.

## 4.4 PENDULUM REGRESSION & MULTISCALE DYNAMICAL SYSTEM PREDICTION

**Pendulum Regression**    Inspired by prior work (Becker et al., 2019; Schirmer et al., 2022), this task involves predicting the sine and cosine of a pendulum's angle from sequences of noisy grayscale images. The input consists of 24x24 pixel images of a pendulum driven by random torque, with added temporally correlated noise. A pendulum is simulated for 100 timesteps, and 50 frames are randomly selected for each sample. We use a dataset split of 2000/1000/1000 samples for training, validation, and testing.

**Multiscale Dynamical System Prediction**    The FitzHugh-Nagumo system (FitzHugh, 1955) models fast-slow nonlinear dynamics simulating neuronal action potentials. Following (Rusch et al., 2022), we approximate this system on sequences of length $N = 1000$, generating multiple datasets

for training, validation, and testing. We compare S7 against various RNN-based models, including the state-of-the-art LEM (Rusch et al., 2022) and S5 (Smith et al., 2023).

| Model | Relative Speed | MSE ($\times 10^{-3}$) |
|---|---|---|
| RKN | $1.9\times$ | 8.43 |
| RKN-$\Delta t$ | $1.9\times$ | 5.09 |
| CRU | $1.0\times$ | 4.63 |
| S5 | $86\times$ | 3.41 |
| **S7 (Ours)** | **$357\times$** | **2.91** |

Table 3: Performance comparison for the Pendulum Regression task. Other models are the same ones as in Smith et al. (2023). S7 achieves the best MSE, outperforming all other models.

| Model | Error ($\times 10^{-2}$) | # Units | # Params |
|---|---|---|---|
| LSTM | 1.2 | 16 | 1k |
| expRNN | 2.3 | 50 | 1k |
| LipschitzRNN | 1.8 | 24 | 1k |
| FastGRNN | 2.2 | 34 | 1k |
| coRNN | 0.4 | 24 | 1k |
| LEM | 0.2 | 16 | 1k |
| S5 | 0.0024 | 16 | 1k |
| **S7 (Ours)** | **0.0013** | 16 | 1k |

Table 4: Test $L^2$ error on FitzHugh-Nagumo system prediction. S7 achieves the best result, outperforming LEM and S5. Other models are the same as in Rusch et al. (2022).

**Results** In the Pendulum Regression task (Table 3), S7 achieves the lowest Mean Squared Error (MSE) of 2.91, outperforming all other models. The large speedup highlights S7's efficiency and ability to handle irregular, noisy inputs. For the Multiscale Dynamical System Prediction task (Table 4), S7 significantly outperforms LEM and S5, achieving the lowest test $L^2$ error of 0.0013. This result demonstrates the advantage of S7's input-dependent recurrent structure for capturing multiscale system dynamics.

## 4.5 Human Activity Recognition & Genomics Classification

**Human Activity Recognition** We evaluate the performance of S7 on the Human Activity Recognition dataset from the UCI repository (Dua & Graff, 2017), a per-time-step classification task involving data collected from four inertial measurement sensors located on a person's arms and feet. Each sensor outputs measurements at fixed intervals of 211 ms, with slight random phase shifts, which introduces irregular sampling in the time-series data. The task is to classify the person's current activity at each time step, making this a challenging sequence modeling problem where every time step presents a new error signal to the network. Other models used for comparison in Table 5 are the same as in Lechner & Hasani (2020).

**Genomics Classification** The EigenWorms dataset (Bagnall et al., 2018) involves classifying worms into either the wild-type or one of four different mutants based on motion data collected over very long sequences. Each sequence has a $N = 17984$ length, making it a challenging task that tests the model's ability to capture very long-term dependencies. Prior research (Rusch et al., 2022; Morrill et al., 2021) has demonstrated that EigenWorms exhibits dependencies that extend beyond 10,000 timesteps, requiring robust sequence modeling techniques to achieve high classification accuracy. Other models in Table 6 are the same as one in Rusch et al. (2022).

| Model | Accuracy (%) |
|---|---|
| ODE-RNN | $80.43 \pm 1.55$ |
| CT-RNN | $83.65 \pm 1.55$ |
| Augmented LSTM | $84.11 \pm 0.68$ |
| CT-GRU | $79.48 \pm 2.12$ |
| RNN Decay | $62.89 \pm 3.87$ |
| Bi-directional RNN | $83.85 \pm 0.45$ |
| GRU-D | $83.57 \pm 0.40$ |
| PhasedLSTM | $83.33 \pm 0.69$ |
| GRU-ODE | $82.56 \pm 2.63$ |
| CT-LSTM | $84.13 \pm 0.11$ |
| ODE-LSTM | $84.15 \pm 0.33$ |
| **S7 (Ours)** | **$94.09 \pm 0.001$** |

Table 5: Per timestep classification. Human Activity Recognition task. Test accuracy (mean $\pm$ std, $N = 5$ experiments for each model).

| Model | Test Accuracy (%) | # Units | # Params |
|---|---|---|---|
| NRDE | 86.8 | 32 | 35k |
| expRNN | 50.1 | 64 | 2.8k |
| IndRNN (2 layers) | 54.5 | 32 | 1.6k |
| LSTM | 48.6 | 32 | 5.3k |
| BiLSTM+1d-conv | 47.8 | 32 | 5.8k |
| chrono-LSTM | 89.0 | 32 | 5.3k |
| coRNN | 89.7 | 32 | 2.4k |
| UniCORNN (2 layers) | 93.3 | 32 | 1.5k |
| LEM | 94.1 | 32 | 5.3k |
| **S7 (Ours)** | **97.5** | 16 | 12k |

Table 6: Test accuracies on EigenWorms using the best-performing models. S7 achieves the best result with the fewest units, demonstrating its effectiveness in capturing long-term dependencies in very long sequences.

| Model | MSE |
|---|---|
| ODE-RNN | $1.904 \pm 0.061$ |
| CT-RNN | $1.198 \pm 0.004$ |
| Augmented LSTM | $1.065 \pm 0.006$ |
| CT-GRU | $1.172 \pm 0.011$ |
| RNN-Decay | $1.406 \pm 0.005$ |
| Bi-directional RNN | $1.071 \pm 0.009$ |
| GRU-D | $1.090 \pm 0.034$ |
| PhasedLSTM | $1.063 \pm 0.010$ |
| GRU-ODE | $1.051 \pm 0.018$ |
| CT-LSTM | $1.014 \pm 0.014$ |
| **S7 (Ours)** | $\mathbf{0.120 \pm 0.005}$ |

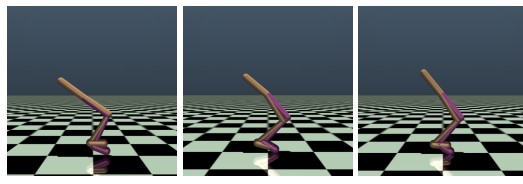

Figure 3: Walker2D kinematic dataset frames visualized.

Figure 2: Per time-step regression results on the Walker2d kinematic dataset. Our S7 model achieves the lowest MSE.

**Results**   In the Human Activity Recognition task (Table 5), S7 achieves a remarkable accuracy of 94.09%, significantly outperforming all baseline models. This substantial improvement underscores S7's ability to effectively handle irregularly sampled, noisy time-series data. For the Genomics Classification task (Table 6), S7 further demonstrates its superiority by attaining a state-of-the-art accuracy of 97.5% using only 16 units and 12k parameters. This result not only surpasses the previous best LEM model (Rusch et al., 2022) but also highlights S7's efficiency in managing very long sequences and capturing long-term dependencies on very challenging dataset with high variance.

### 4.6   WALKER2D KINEMATIC SIMULATION

In this experiment, we evaluated the ability of our proposed S7 model to simulate the kinematic dynamics of the Walker2d-v2 environment. The goal of the task was to predict the per-timestep regression for the kinematic simulation of the MuJoCo physics engine. The dataset was irregularly sampled, and we introduced additional complexity by overwriting a small percentage of the actions with random actions and skipping 10% of the time steps. Table 2 shows that our S7 model achieved the best performance with an MSE of 0.114 (with a mean of 0.120 and std of 0.005), outperforming the other methods by a significant margin. Other models used for comparison are the same ones as in Lechner & Hasani (2020).

## 5   CONCLUSION

In this work, we introduced **S7**, a novel state-space model that effectively balances efficiency, adaptability, and stability in long-sequence modeling tasks—building upon the foundation of prior models like S4 (Gu et al., 2022a), S5 (Smith et al., 2023) and Mamba (Gu & Dao, 2023). S7 leverages input-dependent dynamics and stable reparameterization to improve its ability to capture long-range dependencies while maintaining computational efficiency. The key contribution of S7 is its ability to dynamically adjust state transitions based on input content, allowing for selective filtering and content-based reasoning without adding unnecessary complexity. Through extensive experimentation on a diverse range of benchmarks, including event-based neuromorphic tasks, long-range sequence modeling, dynamical system prediction, and real-world applications like human activity recognition and genomics classification, S7 has demonstrated its superiority. It achieves state-of-the-art results in multiple domains while preserving computational efficiency, even in challenging settings that require processing sequences with irregular sampling and long-term dependencies.

Moreover, incorporating stable reparameterization ensures the robustness and stability of S7 during training and inference, making it highly scalable for real-world applications. The model's ability to handle asynchronous data streams and complex temporal patterns extends its utility to a wide range of practical tasks, from neuromorphic vision to genomic analysis. In conclusion, S7 offers a significant advancement in sequence modeling, pushing the boundaries of what is possible in terms of scalability, generalization, and adaptability. By achieving an optimal balance between simplicity and performance, S7 sets a new standard for state-space models, opening new avenues for research and application across numerous domains in artificial intelligence.

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

# A APPENDIX

## A.1 NOTATIONAL AND THEORETICAL BACKGROUND

To help with the understanding of the theorems, proofs, and the main content of our paper, we provide essential mathematical background on Sobolev norms, properties of functionals, Lipschitz continuity, Grönwall's inequality, and stable reparameterization conditions.

**Sobolev Spaces and Sobolev Norms** Sobolev spaces are a fundamental concept in functional analysis, providing a framework for analyzing functions with weak derivatives. For an open subset $\Omega \subset \mathbb{R}^n$, the Sobolev space $W^{k,p}(\Omega)$ consists of functions whose derivatives up to order $k$ are in $L^p(\Omega)$.

In our context, we consider the Sobolev space $W^{1,\infty}$, which is the space of functions $f : \mathbb{R} \to \mathbb{R}$ such that both $f$ and its first derivative $f'$ are essentially bounded. The Sobolev norm in $W^{1,\infty}$ is defined as:

$$\|f\|_{W^{1,\infty}} = \|f\|_{L^\infty} + \|f'\|_{L^\infty}, \tag{13}$$

where $\|f\|_{L^\infty} = \text{ess sup}_{x \in \Omega} |f(x)|$.

In our analysis, we use the Sobolev-type norm to measure the difference between the target functional $\mathbf{H}$ and the approximate functional $\widehat{\mathbf{H}}$:

$$\|\mathbf{H} - \widehat{\mathbf{H}}\|_{W^{1,\infty}} = \sup_k \left( \|H_k - \widehat{H}_k\|_\infty + \left\| \frac{dH_k}{dk} - \frac{d\widehat{H}_k}{dk} \right\|_\infty \right), \tag{14}$$

where the supremum is taken over all time steps $k$, and the norm $\|\cdot\|_\infty$ denotes the essential supremum over the input space.

**Properties of Functionals** A *functional* is a mapping from a space of functions to the real numbers. In our context, we consider linear functionals $\mathbf{H} : L^\infty(\mathbb{R}) \to \mathbb{R}$ that satisfy the following properties:

- **Boundedness:** The functional $\mathbf{H}$ is bounded if there exists a constant $M > 0$ such that:
$$|\mathbf{H}(u)| \le M\|u\|_{L^\infty}, \quad \forall u \in L^\infty(\mathbb{R}). \tag{15}$$

- **Linearity:** The functional is linear if:
$$\mathbf{H}(au + bv) = a\mathbf{H}(u) + b\mathbf{H}(v), \quad \forall u, v \in L^\infty(\mathbb{R}), \quad \forall a, b \in \mathbb{R}. \tag{16}$$

- **Causality:** The functional is causal if the value at time $k$ depends only on values of $u$ up to time $k$:
$$\mathbf{H}(u)(k) = F(u|_{(-\infty,k]}), \tag{17}$$
where $F$ is some mapping, and $u|_{(-\infty,k]}$ denotes the restriction of $u$ to the interval $(-\infty, k]$.

- **Continuity:** The functional is continuous if small changes in $u$ lead to small changes in $\mathbf{H}(u)$:
$$\lim_{\|u-v\|_{L^\infty} \to 0} |\mathbf{H}(u) - \mathbf{H}(v)| = 0. \tag{18}$$

- **Regularity:** The functional has certain smoothness properties, such as differentiability with respect to $k$.

**Lipschitz Continuity** A function $f : \mathbb{R}^n \to \mathbb{R}^m$ is *Lipschitz continuous* if there exists a constant $L \ge 0$ such that:
$$\|f(x) - f(y)\| \le L\|x - y\|, \quad \forall x, y \in \mathbb{R}^n. \tag{19}$$

The smallest such $L$ is called the *Lipschitz constant* of $f$. Lipschitz continuity ensures that the function does not change too rapidly and is essential for proving stability and convergence results.

In our theorems, we assume that the mappings from the model parameters $\theta_m$ to the system matrices are Lipschitz continuous (Assumption 3.1), which is critical for controlling the effects of parameter perturbations on the system's behavior.

**Grönwall's Inequality**  Grönwall's inequality is a powerful tool used to bound solutions of differential and integral inequalities. It states that if $u(t)$ is a non-negative, continuous function satisfying:

$$u(t) \leq a + \int_{t_0}^{t} b(s)u(s)\, ds, \quad t \geq t_0,$$  (20)

where $a \geq 0$ and $b(s) \geq 0$, then:

$$u(t) \leq a \exp\left( \int_{t_0}^{t} b(s)\, ds \right).$$  (21)

In our proofs, Grönwall's inequality is used to bound the growth of the difference between the perturbed and unperturbed solutions of the state equations, ensuring that small parameter changes lead to proportionally small changes in the system's state over time.

**Stable Reparameterization Conditions**  Reparameterization functions are used to enforce stability constraints on the model parameters. A reparameterization function $f : \mathbb{R} \to \mathbb{R}$ maps raw parameters $w_j$ to model parameters $\theta_m = f(w_m)$. The stability condition requires that the function $f$ ensures the eigenvalues of the state transition matrix $\Lambda_k(u_k; \theta_m)$ have magnitudes less than one (Assumption 3.2).

Moreover, we impose a condition on $f$ that controls the effect of parameter perturbations on the state transition matrices:

$$\sup_{w} \left[ \|f(w)\| \sup_{\|\tilde{w}-w\|\leq\beta} \int_{0}^{\infty} \|\Phi_{\tilde{w}}(k,s) - \Phi_{w}(k,s)\|\ dk \right] \leq g(\beta),$$  (22)

for some continuous function $g : [0, \infty) \to [0, \infty)$ with $g(0) = 0$. This condition ensures that the difference between the perturbed and unperturbed state transition matrices vanishes as the parameter perturbation $\beta \to 0$, promoting stability in the approximation.

**State Transition Matrix**  In systems with time-varying or input-dependent dynamics, the state transition matrix $\Phi(k,s)$ captures the cumulative effect of the state transition matrices from time $s$ to $k$. It satisfies the difference equation:

$$\Phi(k,s) = \Lambda_k(u_k; \theta_m)\Phi(k-1,s), \quad \Phi(s,s) = I_m,$$  (23)

where $I_m$ is the identity matrix of size $m$. The state transition matrix is crucial for expressing the solution to the state equation and analyzing the system's behavior over time.

**Variation of Parameters**  The variation of parameters is a method for solving non-homogeneous linear differential or difference equations. For the difference equation:

$$x_k = \Lambda_k x_{k-1} + B_k u_k + b_k,$$  (24)

the solution can be expressed as:

$$x_k = \Phi(k,k_0)x_{k_0} + \sum_{s=k_0+1}^{k} \Phi(k,s)(B_s u_s + b_s),$$  (25)

where $\Phi(k,s)$ is the state transition matrix. This representation allows us to analyze how inputs and initial conditions influence the system's state.

**Bounding the Approximation Error**  In the context of our theorems, we aim to show that the sequence of approximate functionals $\{\widehat{\mathbf{H}}(\cdot; \theta_m)\}_{m=1}^{\infty}$ converges to the target functional $\mathbf{H}$ in the Sobolev norm. The total approximation error $E(\beta)$ combines the error due to model capacity (which decreases as $m \to \infty$) and the error from parameter perturbations (controlled by $\beta$). By ensuring that both errors tend to zero, we establish that the approximation is stable and accurate.

**Gradient Norm Scaling**   In Theorem 3.6, we analyze how the gradient of the loss function with respect to the raw parameters $w_j$ scales with the derivative of the reparameterization function $f$. The key result is that:

$$\left| \frac{\partial \text{Loss}}{\partial w_j} \right| \leq C_{\mathbf{H},\widehat{\mathbf{H}}_m} \left| f'(w_j) \right|, \tag{26}$$

where $C_{\mathbf{H},\widehat{\mathbf{H}}_m}$ is a constant independent of $w_j$. This highlights the importance of choosing a reparameterization function with appropriate smoothness to ensure that gradients are well-behaved during optimization.

**Ensuring Stability and Convergence**   Combining the above concepts, our analysis ensures that the input-dependent state-space models we propose are both stable and capable of providing accurate approximations of target functionals. The Lipschitz continuity and stability conditions prevent the system from exhibiting uncontrolled behavior, while the use of Sobolev norms allows us to measure approximation quality in terms of both the function value and its derivative.

The theoretical results provide a solid foundation for the practical effectiveness of our S7 model. By ensuring stability and controlling gradient norms, we can train deep models capable of handling long sequences with complex dependencies. The input-dependent dynamics enable the model to adapt to varying inputs, improving its ability to capture long-range dependencies and perform content-based reasoning without sacrificing computational efficiency.

## A.2   PROOF OF THEOREM 3.5

In this section, we prove the Theorem regarding the Existence of Stable Approximation by Stable Reparameterization with Input-Dependent Dynamics.

*Proof.* We begin by defining the target linear functional $\mathbf{H}$ as follows:

$$H_k(\mathbf{u}) = \int_{-\infty}^{k} \rho(k - s) u_s \, ds, \tag{27}$$

where $\rho$ is an $L_1$-integrable function, meaning $\int_0^{\infty} |\rho(\tau)| \, d\tau < \infty$. The objective is to approximate $\mathbf{H}$ using a sequence of state-space models with input-dependent dynamics. The modified state-space model takes the form:

$$\frac{dx_k}{dk} = \Lambda_k(u_k) x_k + B u_k + b, \tag{28}$$

with the output defined as $\hat{y}_k = c^\top x_k$, where $c \in \mathbb{R}^m$ is the output weight vector. The key difference from prior models lies in the input dependence of $\Lambda_k(u_k)$, which makes the state equation non-autonomous. This complexity implies that the solution to the state equation cannot simply be expressed using the exponential of a constant matrix. The solution $x_k$ to the state equation involves the state transition matrix, which depends on the history of $u_k$:

$$x_k = \Phi(k, k_0) x_{k_0} + \int_{k_0}^{k} \Phi(k, s)(B u_s + b) \, ds, \tag{29}$$

where $\Phi(k, s)$ is the state transition matrix from time step $s$ to $k$, satisfying $\frac{d}{dk}\Phi(k, s) = \Lambda_k(u_k)\Phi(k, s)$ with $\Phi(s, s) = I_m$. Since $\Lambda(u_k)$ depends on $u_k$, $\Phi(k, s)$ depends on the entire input sequence $u_{[s,k]}$.

We approximate the state transition matrix using a piecewise constant approximation of $\Lambda_k(u_k)$. This means we divide the interval $[k_0, k]$ into small subintervals $[k_{i-1}, k_i]$ where $\Lambda_k(u_k)$ is approximately constant, allowing us to express the state transition matrix as $\Phi(k, k_0) \approx \prod_{i=1}^{N} e^{\Lambda_k(u_{k_{i-1}})(k_i - k_{i-1})}$. This approximation becomes more accurate as the intervals become smaller. The model output is given by $\hat{y}_k = c^\top x_k$, while the target functional is $H_k(\mathbf{u}) = \int_{-\infty}^{k} \rho(k - s) u_s \, ds$. Our goal is to show that $\hat{y}_k$ approximates $H_k(\mathbf{u})$ under appropriate conditions. The total approximation error $E(\beta)$ can be expressed as:

$$E(\beta) = \sup_{|\tilde{\theta} - \theta| \leq \beta} \|\mathbf{H} - \widehat{\mathbf{H}}(\cdot; \tilde{\theta})\|_{W^{1,\infty}}. \tag{30}$$

where $\theta$ represents the model parameters, and $\tilde{\theta}$ represents the perturbed parameters within a radius $\beta$. We need to bound $E(\beta)$ and show that $\lim_{\beta \to 0} E(\beta) = 0$.

Perturbations in $\theta$ affect both $\Lambda_k(u_k)$ and the state transition matrix $\Phi(k, s)$, but if $\Lambda_k(u_k; \theta)$ depends smoothly on $\theta$ and the mapping from $\theta$ to $\Lambda_k(u_k; \theta)$ is Lipschitz continuous, then small perturbations in $\theta$ yield small perturbations in the system dynamics.

To analyze the difference between the perturbed and unperturbed state transition matrices, consider $\Phi(k, s)$ for the unperturbed case and $\tilde{\Phi}(k, s)$ for the perturbed case. We seek to bound $\|\tilde{\Phi}(k, s) - \Phi(k, s)\|$. Assuming $\Lambda_k(u_k; \theta)$ is Lipschitz in $\theta$, and that $u_k$ is bounded, we can establish that:

$$\|\tilde{\Phi}(k, s) - \Phi(k, s)\| \le L_\Phi \beta (k - s), \tag{31}$$

for some constant $L_\Phi$, where $\beta = \|\tilde{\theta} - \theta\|$. This gives us a first step in bounding the overall approximation error. The error in the output can now be written as $|\hat{y}_k - H_k(\mathbf{u})| = |c^\top (x_k - x_k^{\text{target}})|$, where $x_k^{\text{target}}$ corresponds to the hidden state that exactly reproduces $H_k(\mathbf{u})$. The difference $x_k - x_k^{\text{target}}$ arises from two sources: the model approximation error and the perturbation in parameters. We express this as:

$$|\hat{y}_k - H_k(\mathbf{u})| \le \|c\| \cdot \|x_k - x_k^{\text{target}}\|. \tag{32}$$

To control the error due to parameter perturbations, we analyze the difference between the hidden states $\tilde{x}_k$ (with perturbed parameters $\tilde{\theta}$) and $x_k$ (with original parameters $\theta$). The difference $\delta x_k = \tilde{x}_k - x_k$ satisfies the following differential equation:

$$\frac{d \delta x_k}{dk} = \Lambda_k(u_k; \tilde{\theta}) \delta x_k + [\Lambda_k(u_k; \tilde{\theta}) - \Lambda_k(u_k; \theta)] x_k + [B(\tilde{\theta}) - B(\theta)] u_k + [b(\tilde{\theta}) - b(\theta)]. \tag{33}$$

Using the Lipschitz continuity of $\Lambda_k(u_k; \theta)$, $B(\theta)$, and $b(\theta)$, we know that:

$$\|\Lambda_k(u_k; \tilde{\theta}) - \Lambda_k(u_k; \theta)\| \le L_\Lambda \beta, \tag{34}$$

with similar bounds for $B$ and $b$. Applying Grönwall's inequality, we bound the growth of $\delta x_k$ as:

$$\|\delta x_k\| \le \int_{k_0}^k e^{L(k-s)} \left( L_\Lambda \|x_s\| \beta + L_B \|u_s\| \beta + L_b \beta \right) ds. \tag{35}$$

Given that both $x_s$ and $u_s$ are bounded and that $k - s$ remains finite, the integral yields a bound of the form $\|\delta x_k\| \le C\beta$, where $C$ is a constant depending on the system's bounds.

Finally, this leads to the bound on the output error:

$$|\hat{y}_k(\tilde{\theta}) - \hat{y}_k(\theta)| = |c^\top (\tilde{x}_k - x_k)| \le \|c\| \cdot \|\delta x_k\| \le \|c\| C\beta. \tag{36}$$

Thus, the total approximation error $E(\beta)$ satisfies:

$$E(\beta) \le E(0) + K\beta, \tag{37}$$

where $E(0) \to 0$ as $m \to \infty$ and $K$ is a constant. Therefore, $\lim_{\beta \to 0} E(\beta) = 0$, demonstrating that the approximation is stable as the model size grows (sequence of state-space models provides a stable approximation of the target functional). $\qquad \square$

### A.3 PROOF OF THEOREM 3.6

*Proof.* We aim to establish that under the given assumptions, the gradient of the loss function with respect to the trainable parameter $w_j$ is bounded by:

$$\left| \frac{\partial \text{Loss}}{\partial w_j} \right| \le C_{\mathbf{H}, \widehat{\mathbf{H}}_m} |f'(w_j)|, \tag{38}$$

where $C_{\mathbf{H}, \widehat{\mathbf{H}}_m}$ is a constant independent of $w_j$.

Consider the loss function defined as:

$$\text{Loss} = \sup_k \|H_k(\mathbf{u}) - \hat{y}_k(\mathbf{u})\|_\infty, \tag{39}$$

where $H_k(\mathbf{u})$ is the target functional, and $\hat{y}_k(\mathbf{u})$ is the model output at time step $k$ given input $\mathbf{u}$. Since the loss involves a supremum over inputs $\mathbf{u}$ with $\|\mathbf{u}\|_\infty \leq 1$, we focus on bounding the gradient of $\hat{y}_k(\mathbf{u})$ with respect to $w_j$.

The gradient of the loss with respect to $w_j$ is given by:

$$\left| \frac{\partial \mathrm{Loss}}{\partial w_j} \right| = \left| \frac{\partial}{\partial w_j} \sup_{\|\mathbf{u}\|_\infty \leq 1} \|H_k(\mathbf{u}) - \hat{y}_k(\mathbf{u})\|_\infty \right|. \tag{40}$$

Noting that $H_k(\mathbf{u})$ does not depend on $w_j$, we can write:

$$\left| \frac{\partial \mathrm{Loss}}{\partial w_j} \right| \leq \sup_{\|\mathbf{u}\|_\infty \leq 1} \left| \frac{\partial \hat{y}_k(\mathbf{u})}{\partial w_j} \right|. \tag{41}$$

Our goal is thus to bound $\left| \frac{\partial \hat{y}_k(\mathbf{u})}{\partial w_j} \right|$. The model output is defined as:

$$\hat{y}_k = c(\theta_m)^\top x_k, \tag{42}$$

where $c(\theta_m) \in \mathbb{R}^m$ is a parameter-dependent vector, and $x_k \in \mathbb{R}^m$ is the hidden state at time step $k$. Taking the derivative of $\hat{y}_k$ with respect to $w_j$, we have:

$$\frac{\partial \hat{y}_k}{\partial w_j} = \left( \frac{\partial c(\theta_m)}{\partial w_j} \right)^\top x_k + c(\theta_m)^\top \frac{\partial x_k}{\partial w_j}.$$

The first term involves the derivative of $c(\theta_m)$, and the second term consists of the derivative of the hidden state $x_k$.

Since $c(\theta_m)$ is Lipschitz continuous with respect to $\theta_m$ (Assumption 3.1), and $\theta_m = f(w_m)$, where $w_m$ is the vector of trainable parameters, we can bound the first term using the chain rule:

$$\left\| \frac{\partial c(\theta_m)}{\partial w_j} \right\| = \left\| \frac{\partial c(\theta_m)}{\partial \theta_m} \cdot \frac{\partial \theta_m}{\partial w_j} \right\| \leq L_c \left\| \frac{\partial \theta_m}{\partial w_j} \right\| = L_c |f'(w_j)|, \tag{43}$$

where $L_c$ is the Lipschitz constant of $c$ with respect to $\theta_m$, and $f'(w_j)$ is the derivative of the reparameterization function $f$ with respect to $w_j$.

To bound the second term, we need to compute $\delta x_k^j := \frac{\partial x_k}{\partial w_j}$. Differentiating the state equation with respect to $w_j$, we obtain:

$$\frac{d \delta x_k^j}{dk} = \Lambda_k(u_k; \theta_m) \delta x_k^j + \left( \frac{\partial \Lambda_k(u_k; \theta_m)}{\partial w_j} \right) x_k + \left( \frac{\partial B(\theta_m)}{\partial w_j} \right) u_k + \frac{\partial b(\theta_m)}{\partial w_j}. \tag{44}$$

This is a non-homogeneous linear difference equation for $\delta x_k^j$.

Using the chain rule and the Lipschitz continuity of $\Lambda_k(u_k; \theta_m)$, $B(\theta_m)$, and $b(\theta_m)$ with respect to $\theta_m$ (Assumption 3.1), we have:

$$\left\| \frac{\partial \Lambda_k(u_k; \theta_m)}{\partial w_j} \right\| \leq L_\Lambda |f'(w_j)|, \quad \left\| \frac{\partial B(\theta_m)}{\partial w_j} \right\| \leq L_B |f'(w_j)|, \quad \left\| \frac{\partial b(\theta_m)}{\partial w_j} \right\| \leq L_b |f'(w_j)|, \tag{45}$$

where $L_\Lambda$, $L_B$, and $L_b$ are the Lipschitz constants of $\Lambda_k$, $B$, and $b$ with respect to $\theta_m$, respectively.

The solution to the difference equation for $\delta x_k^j$ can be expressed using the variation of parameters formula:

$$\delta x_k^j = \int_{k_0}^k \Phi(k, s) \left( \left( \frac{\partial \Lambda_k(u_s; \theta_m)}{\partial w_j} \right) x_s + \left( \frac{\partial B(\theta_m)}{\partial w_j} \right) u_s + \frac{\partial b(\theta_m)}{\partial w_j} \right) ds, \tag{46}$$

where $\Phi(k, s)$ is the state transition matrix given by:

$$\Phi(k, s) = \mathcal{T} \exp \left( \int_s^k \Lambda_k(u_\tau; \theta_m) d\tau \right), \tag{47}$$

and $\mathcal{T}$ denotes the time-ordering operator.

Under Assumption 3.2, the system is uniformly asymptotically stable; thus, there exist constants $M > 0$ and $\alpha > 0$ such that:

$$\|\Phi(k, s)\| \leq Me^{-\alpha(k-s)}, \quad \text{for all } k \geq s. \tag{48}$$

This property ensures that the effect of the initial conditions and perturbations diminishes exponentially over time. Since the hidden states $x_s$ and inputs $u_s$ are uniformly bounded (Assumption 3.4), there exist constants $K_x$ and $K_u$ such that:

$$\|x_s\| \leq K_x, \quad \|u_s\| \leq K_u, \quad \text{for all } s. \tag{49}$$

Substituting the bounds into the expression for $\delta x_k^j$, we have:

$$\|\delta x_k^j\| \leq \int_{k_0}^k \|\Phi(k, s)\| \left( L_\Lambda |f'(w_j)| K_x + L_B |f'(w_j)| K_u + L_b |f'(w_j)| \right) ds. \tag{50}$$

Simplifying, we obtain:

$$\|\delta x_k^j\| \leq \left( L_\Lambda K_x + L_B K_u + L_b \right) |f'(w_j)| \int_{k_0}^k Me^{-\alpha(k-s)} ds. \tag{51}$$

Evaluating the integral, we find:

$$\int_{k_0}^k Me^{-\alpha(k-s)} ds = \frac{M}{\alpha} \left( 1 - e^{-\alpha(k-k_0)} \right) \leq \frac{M}{\alpha}. \tag{52}$$

Therefore, the bound on $\|\delta x_k^j\|$ becomes:

$$\|\delta x_k^j\| \leq \frac{M}{\alpha} \left( L_\Lambda K_x + L_B K_u + L_b \right) |f'(w_j)| = C_x |f'(w_j)|, \tag{53}$$

where $C_x = \frac{M}{\alpha} \left( L_\Lambda K_x + L_B K_u + L_b \right)$ is a constant independent of $w_j$.

Returning to the expression for $\frac{\partial \hat{y}_k}{\partial w_j}$, we can now bound each term. The first term satisfies:

$$\left\| \left( \frac{\partial c(\theta_m)}{\partial w_j} \right)^\top x_k \right\| \leq \left\| \frac{\partial c(\theta_m)}{\partial w_j} \right\| \|x_k\| \leq L_c |f'(w_j)| K_x. \tag{54}$$

The second term satisfies:

$$\left\| c(\theta_m)^\top \delta x_k^j \right\| \leq \|c(\theta_m)\| \|\delta x_k^j\| \leq K_c C_x |f'(w_j)|, \tag{55}$$

where $K_c = \|c(\theta_m)\|$ is uniformly bounded (from Assumption 3.4). Combining these bounds, we have:

$$\left| \frac{\partial \hat{y}_k}{\partial w_j} \right| \leq \left( L_c K_x + K_c C_x \right) |f'(w_j)| = C_y |f'(w_j)|, \tag{56}$$

where $C_y = L_c K_x + K_c C_x$ is a constant independent of $w_j$.

Finally, substituting back into the bound for the gradient of the loss, we obtain:

$$\left| \frac{\partial \text{Loss}}{\partial w_j} \right| \leq \sup_{\|\mathbf{u}\|_\infty \leq 1} \left| \frac{\partial \hat{y}_k(\mathbf{u})}{\partial w_j} \right| \leq C_y |f'(w_j)|. \tag{57}$$

Thus, the gradient of the loss with respect to $w_j$ is bounded by:

$$\left| \frac{\partial \text{Loss}}{\partial w_j} \right| \leq C_{\mathbf{H}, \widehat{\mathbf{H}}_m} |f'(w_j)|, \tag{58}$$

where $C_{\mathbf{H}, \widehat{\mathbf{H}}_m} = C_y$ depends on the model parameters and the target functional but is independent of $w_j$.

This completes the proof, demonstrating that in input-dependent state-space models, the gradient norm with respect to the trainable parameters $w_j$ is directly proportional to $|f'(w_j)|$. The constants involved in the bound are determined by the Lipschitz constants of the system components, the bounds on the hidden states and inputs, and the stability properties of the system, all of which are independent of $w_j$. This highlights the critical role of the reparameterization function $f$ in controlling gradient scales during optimization. Thus, the appropriate choice of $f$ is essential for stable and efficient training in models with input-dependent dynamics. $\qquad\square$

### A.4 CHOOSING THE RIGHT REPARAMETERIZATION

In our model, the reparameterization function $f$ is crucial in ensuring stability during training and controlling the gradient norms. According to Theorem 3.6, the gradient of the loss with respect to the raw parameter $w_j$ scales with the *magnitude* of the derivative of the reparameterization function $f$:

$$\left|\frac{\partial \text{Loss}}{\partial w_j}\right| \leq C_{\mathbf{H},\widehat{\mathbf{H}}_m} |f'(w_j)|. \tag{59}$$

To promote stable and efficient training, it is desirable for the gradient magnitude to be proportional to the parameter magnitude, i.e.,

$$\left|\frac{\partial \text{Loss}}{\partial w_j}\right| \leq L|w_j|, \tag{60}$$

for some constant $L > 0$. Combining this with equation (59), we obtain the following condition on the reparameterization function:

$$C_{\mathbf{H},\widehat{\mathbf{H}}_m} |f'(w_j)| \leq L|w_j|. \tag{61}$$

Our aim is to find a reparameterization function $f$ satisfying this condition. Rearranging, we get:

$$|f'(w)| \leq \frac{L}{C_{\mathbf{H},\widehat{\mathbf{H}}_m}} |w|. \tag{62}$$

This differential inequality suggests that $f$ should be such that its derivative $f'(w)$ is proportional to $w$. However, to ensure the stability of the system and to control the gradient norms effectively, we consider a more refined condition based on the relationship between $f$, $f'$, and the parameter $w$.

Suppose we define the function $G_f(w)$ as:

$$G_f(w) = \frac{|f'(w)|}{f(w)^2}. \tag{63}$$

Our goal is to find $f$ such that:

$$G_f(w) = \frac{|f'(w)|}{f(w)^2} = L|w|, \tag{64}$$

for some constant $L > 0$. This condition arises from the consideration that the gradient-over-weight ratio should be bounded, which is crucial for training stability.

Solving the differential equation (64), we integrate both sides:

$$\frac{f'(w)}{f(w)^2} = 2aw, \quad \text{where } a = \frac{L}{2}, \tag{65}$$

$$\Rightarrow \int \frac{f'(w)}{f(w)^2} \, dw = \int 2aw \, dw, \tag{66}$$

$$\Rightarrow -\frac{1}{f(w)} = aw^2 + b, \tag{67}$$

where $b$ is the constant of integration. Therefore, the reparameterization function is:

$$f(w) = -\frac{1}{aw^2 + b}. \tag{68}$$

To ensure stability, we require that $f(w) \leq 0$ for all $w$. Moreover, in the discrete case relevant to our model, we can consider:

$$f(w) = 1 - \frac{1}{aw^2 + b}. \tag{69}$$

By choosing appropriate values for $a$ and $b$, we can ensure that the reparameterization function $f(w)$ satisfies the stability conditions and promotes a bounded gradient-over-weight ratio. In our experiments, we set $a = 1$ and $b = 0.5$, which ensures that $f(w)$ remains within the stability region and that $f(w)$ does not cross critical boundaries (e.g., for eigenvalues in recurrent models).

**Ablation Study on Reparameterization Choices**   In our ablation study (see Appendix A.6), we experiment with different choices of the constants $a$ and $b$ in the reparameterization function (69). We find that adjusting these parameters affects the stability and performance of the model. Specifically, the reparameterization with $a = 1$ and $b = 0.5$ consistently outperforms other choices, providing the best balance between stability and performance.

Moreover, we compare models trained with and without reparameterization. The models with reparameterization achieve better performance because they exhibit more stable training dynamics. This demonstrates the effectiveness of the reparameterization strategy in improving both the stability and the performance of the S7 model.

**Remarks**   While gradient clipping is a common technique to prevent exploding gradients, it can introduce bias and reduce the effectiveness of gradient descent. In contrast, our reparameterization approach inherently controls the gradient scales by modifying the parameterization of the model. This acts as a form of preconditioning, improving optimization without the drawbacks associated with gradient clipping. Our findings highlight the importance of choosing an appropriate reparameterization function to ensure stable and efficient training. The "best" reparameterization derived from equation (68) offers a theoretically grounded and empirically validated approach to achieving this goal.

## A.5   DETAILS OF LONG RANGE ARENA TASKS

**ListOps**   Tests a model's ability to compute nested mathematical expressions with sequences of up to 2,000 tokens. S7 outperforms both Mega (Ma et al., 2023) and S5 (Smith et al., 2023), achieving a score of 63.77. S7's input-dependence mechanism aids in filtering repetitive tokens and maintaining logical consistency, contributing to its superior performance in structured reasoning tasks.

**Text**   This task involves classifying IMDb movie reviews as positive or negative, with sequences padded to 4,096 tokens. S7 scores 87.22, having a very close performance to the S5 (Smith et al., 2023) and Mega (Ma et al., 2023) models.

**Retrieval**   In this task, models determine if two textual citations are equivalent, with sequences of up to 4,000 tokens. S7 achieves the highest score of 91.80, indicating that its dynamic state-space architecture is well-suited for tasks requiring long-range memory and retrieval capabilities.

**Image**   This task involves classifying CIFAR-10 images as 1D raster scans of 1,024 tokens. S7 performs significantly worse with input-dependence, scoring 61.14 compared to Mega's (Ma et al., 2023) 90.44 and S5's (Smith et al., 2023) 88.00. This suggests that input-dependence disrupts spatial reasoning by inadvertently discarding crucial tokens, leading to information loss in tasks where maintaining a precise spatial structure, like in image classification, is essential for accurate predictions.

**Pathfinder**   Pathfinder is a binary classification task where models predict if a path in a maze-like image connects two points. S7's performance drops significantly to 65.52% with input-dependence enabled, compared to S5's 95.33%, indicating that input-dependence negatively impacts tasks requiring precise spatial reasoning. This suggests that simpler models without input-dependence are more effective for visuospatial tasks where retaining exact input information is crucial.

**Path-X**   This is the most challenging task with sequences of 16,384 tokens and requires models to identify long-range visual patterns. S7 achieves a score of 61.50%, indicating a significant drop in performance with input-dependence enabled compared to models like S5. This suggests that input-dependent dynamics can hinder performance in tasks requiring precise retention of input information over very long sequences, as the forgetting mechanism introduced by input dependence leads to the loss of crucial spatial details necessary for accurate classification.

## A.6 ABLATION STUDY

### A.6.1 IMPORTANCE OF REPARAMETERIZATION

In this section, we conduct an ablation study to assess the impact of including stable reparameterization on the performance of the S7 model across various datasets. We compare models trained with and without the reparameterization, as discussed in Section A.4. The datasets used in this study include DVS-Gesture (Amir et al., 2017), Spiking Heidelberg Digits (SHD) (Cramer et al., 2019), Spiking Speech Commands (SSC) (Cramer et al., 2019), Human Activity Recognition (Dua & Graff, 2017), EigenWorms (Bagnall et al., 2018), and several tasks from the Long Range Arena (LRA) benchmark (Tay et al., 2021a).

The results are presented in Tables 7, 8, and 9.

| Dataset | Reparameterization | Accuracy (%) |
|---------|--------------------|--------------|
| DVS-Gesture | No | 98.1 |
|             | Yes | **99.2** |
| SHD | No | 93.1 |
|     | Yes | **96.3** |
| SSC | No | 87.8 |
|     | Yes | **88.2** |

Table 7: Ablation study on event-based datasets: comparison of S7 model performance with and without stable reparameterization.

| Dataset | Reparameterization | Accuracy (%) |
|---------|--------------------|--------------|
| Human Activity | No | 93.79 |
|                | Yes | **94.09** |
| EigenWorms | No | 96.66 |
|            | Yes | **97.50** |

Table 8: Reparametrization ablation study on Human Activity Recognition and EigenWorms datasets.

| LRA Task | Reparameterization | Accuracy (%) |
|----------|--------------------|--------------|
| ListOps | No | 62.11 |
|         | Yes | **63.77** |
| Text | No | 85.42 |
|      | Yes | **87.22** |
| Retrieval | No | 91.64 |
|           | Yes | **91.80** |
| Image | No | 60.30 |
|       | Yes | **61.14** |

Table 9: Reparametrization Ablation study on LRA tasks.

**Analysis of Results** From the results presented in Tables 7, 8, and 9, it is evident that incorporating stable reparameterization consistently improves the performance of the S7 model across all considered datasets.

In the **event-based datasets** (Table 7), the inclusion of reparameterization leads to significant accuracy gains. On the DVS-Gesture dataset, accuracy improves from 98.1% without reparameterization to 99.2% with reparameterization. For the Spiking Heidelberg Digits, accuracy increases from 93.1% to 96.3%, and on the Spiking Speech Commands dataset, the model with reparameterization achieves 87.8% accuracy compared to 88.2% without it.

In the **Human Activity Recognition** and **EigenWorms** datasets (Table 8), similar improvements are observed. The Human Activity Recognition task sees an accuracy rise from 93.79% to 94.09% with reparameterization. For the EigenWorms dataset, accuracy increases from 96.66% to 97.50%, highlighting the model's enhanced ability to capture long-range dependencies in very long sequences.

In the **Long Range Arena tasks** (Table 9), the models with reparameterization outperform those without across all tasks. On the ListOps task, accuracy improves from 62.11% to 63.77%. For the Text classification task, accuracy increases from 85.42% to 87.22%. In the Retrieval task, the model achieves 91.80% accuracy with reparameterization, compared to 91.64% without. On the Image classification task, accuracy improves from 60.30% to 61.14%.

These consistent performance gains suggest that the inclusion of stable reparameterization improves the S7 model's ability to learn effectively from diverse types of data. The reparameterization contributes to more controlled gradient norms and improved training stability, allowing the model to better capture complex temporal patterns and long-range dependencies.

### A.6.2 Effect of Reparameterization Parameters

We also explore the impact of different choices for the parameters $a$ and $b$ in the reparameterization function $f(w) = 1 - \frac{1}{aw^2+b}$, as discussed in Section A.4. By conducting experiments varying these parameters, we aim to identify the configuration that yields the best performance.

| Dataset | $a = 0.5, b = 0.5$ | $a = 1, b = 0.5$ | $a = 1, b = 1$ |
|---|---|---|---|
| DVS-Gesture (Amir et al., 2017) | 98.7% | **99.2%** | 98.9% |
| EigenWorms (Bagnall et al., 2018) | 96.8% | **97.5%** | 97.1% |
| ListOps (Tay et al., 2021a) | 62.53% | **63.77%** | 63.22% |

Table 10: Effect of different reparameterization parameters $a$ and $b$ on model performance.

As shown in Table 10, setting $a = 1$ and $b = 0.5$ consistently yields the best performance across datasets. This configuration effectively balances stability and gradient scaling, providing controlled gradient norms without adversely affecting the model's expressiveness.

**Conclusion** The ablation studies confirm that the stable reparameterization is crucial for the S7 model's performance and training stability. By carefully choosing the reparameterization parameters, specifically $a = 1$ and $b = 0.5$, we achieve optimal results across various tasks.

The improvements observed across diverse datasets, including event-based data, human activity recognition, genomics classification, and long-range sequence tasks, show the generality and robustness of the reparameterization approach. Incorporating stable reparameterization not only improves performance but also contributes to more stable training dynamics, enabling the S7 model to better capture long-range dependencies and complex temporal patterns inherent in sequential data.

### A.7 Best Hyperparameters

In this section, we provide the hyperparameters used for training the best-performing S7 models across various tasks. The hyperparameters are summarized in Table 11. The tasks include event-based datasets, long-range sequence modeling benchmarks, and other sequence classification tasks. The hyperparameters were carefully selected through Bayesian search and experimentation to optimize model performance while ensuring training stability.

| Task | Depth | H | Dropout | P | J | LR | SSM LR | WD | Epochs | B |
|---|---|---|---|---|---|---|---|---|---|---|
| **DVS-Gesture** | 6 | 32 | 0.10 | 16 | 1 | $1.2 \times 10^{-5}$ | $1.44 \times 10^{-4}$ | 0.000 | 100 | 3 |
| **EigenWorms** | 1 | 16 | 0.03 | 14 | 7 | $5.6 \times 10^{-5}$ | $6.78 \times 10^{-4}$ | 0.044 | 900 | 12 |
| **Image (LRA)** | 2 | 60 | 0.15 | 24 | 12 | $1.0 \times 10^{-5}$ | $2.79 \times 10^{-3}$ | 0.015 | 200 | 280 |
| **ListOps (LRA)** | 6 | 102 | 0.23 | 8 | 1 | $5.0 \times 10^{-6}$ | $3.42 \times 10^{-4}$ | 0.065 | 200 | 64 |
| **Pathfinder (LRA)** | 6 | 50 | 0.00 | 2 | 1 | $5.0 \times 10^{-5}$ | $9.23 \times 10^{-4}$ | 0.010 | 200 | 16 |
| **Human Activity Recognition** | 1 | 120 | 0.04 | 64 | 32 | $9.3 \times 10^{-5}$ | $7.42 \times 10^{-3}$ | 0.019 | 400 | 80 |
| **Retrieval (LRA)** | 3 | 80 | 0.10 | 10 | 2 | $2.8 \times 10^{-5}$ | $5.04 \times 10^{-4}$ | 0.045 | 90 | 18 |
| **Spiking Heidelberg Digits** | 6 | 48 | 0.14 | 8 | 1 | $4.8 \times 10^{-5}$ | $1.54 \times 10^{-3}$ | 0.021 | 30 | 32 |
| **Spiking Speech Commands** | 8 | 32 | 0.25 | 32 | 4 | $1.1 \times 10^{-5}$ | $8.68 \times 10^{-5}$ | 0.004 | 200 | 8 |
| **Text (LRA)** | 6 | 96 | 0.31 | 4 | 1 | $3.2 \times 10^{-5}$ | $1.03 \times 10^{-3}$ | 0.021 | 200 | 32 |
| **Walker2d** | 1 | 100 | 0.21 | 32 | 16 | $3.7 \times 10^{-5}$ | $2.25 \times 10^{-3}$ | 0.085 | 100 | 60 |

Table 11: Hyperparameters used for training the best S7 models. Depth: number of layers. H: number of input/output features. P: latent size. J: number of blocks used for the initialization of $\Lambda$. LR: base learning rate. SSM LR: learning rate for SSM parameters. WD: weight decay. Epochs: number of training epochs. B: batch size.

