# OpenReview forum: "S7: Selective and Simplified State Space Layers for Sequence Modeling"
_ICLR.cc/2025/Conference — ICLR 2025 Conference Withdrawn Submission_

### Official Review · Reviewer_s9V1 · 2024-11-01

**Soundness:** 2
**Presentation:** 2
**Contribution:** 1
**Rating:** 3
**Confidence:** 5

**Summary:**

The paper proposes a novel state-space model (SSM) called S7 that introduces input-dependent dynamics to filter and retain relevant input information over extended time horizons. An important part of S7 is a reparametrization trick that ensures training stability. S7 is claimed to reach a balance between performance and model's complexity for processing very long sequences.

**Strengths:**

- The authors evaluate their model on a series of benchmark tasks
- The authors provide some theoretical analysis for the training stability

**Weaknesses:**

- Scientific novelty of the manuscript: the authors base their work on the existing S5 model and extend it with the input-gating, that other models, such as Mamba already have demonstrated. In fact, it remains even unclear when reading through the main paper how the input-gating is realized precisely. The reader misses those equations. I.e., How is Lambda_k computed?
- The selection of the tasks and in particular the selection of the reference models is a major weakness of the model:
    1. since the S7 model is based on the S5 model, it is of paramount importance that one always compares to S5 at least, which the authors do not do for many datasets they considered. For example, this comparison is missing in Table 5, 6 and 7 (wrongly called Figure 2 in the manuscript).
    2. looking at the LRA results, one can see that S7 is only in 2 tasks slightly better than S5, but in the remaining tasks it is significantly worse. Moreover, in Table 5, 6 and 7 (wrongly called Figure 2 in the manuscript), the authors don’t even compare to the S5.
the authors claim to introduce a simpler model than Mambda, but it remains unclear in what regards it is simpler, e.g., if it uses less number of parameters, or less computations, this needs to be demonstrated in the results.
- Some minor comments are: The authors are very much overstating their novel contributions with terms such as “S7 has demonstrated its superiority”, which is by no means true. Figure 2 on page 10 should probably be Table 7

**Questions:**

- Is the code to reproduce the results publicly available?
- How is lambda_k computed?
- Fig.1 delta_k is not defined in the caption
- Line 104-109 this sentence is too long and could be split in 2-3 smaller ones.
- What is the performance of other SSMs in Table 5, 6 and 7?
- In which regards in the S7 simpler or superior over other SSMs? Number of flops? Number of parameters?

---

### Official Review · Reviewer_wghn · 2024-11-01

**Soundness:** 2
**Presentation:** 1
**Contribution:** 2
**Rating:** 3
**Confidence:** 5

**Summary:**

The paper presents S7, a simplified state-space model (SSM) designed for long-range sequence tasks. Building on the S5 model, it introduces input-dependence to allow dynamic adjustments to state transitions based on input content. The paper claims S7 achieves stable reparameterization and efficient performance across diverse tasks, including neuromorphic event-based datasets and Long Range Arena (LRA) benchmarks, without requiring the complexity of models like S4 or S6 (Mamba). The proposed model is argued to balance computational simplicity with adaptive filtering and content-based reasoning.

**Strengths:**

*Input-Dependent Dynamics:* S7’s adaptation of the  S5 model to be input-dependent is a promising approach. This could enhance the model’s responsiveness to input variability, a significant issue in long-range sequence tasks.

*Stable Reparameterization:* The model claims to maintain gradient stability over long sequences, addressing gradient explosion/vanishing issues commonly faced in deep learning. This feature has potential benefits for training efficiency and stability.

*Broad Applicability:* S7’s successful application across various domains, from physical time series to neuromorphic vision, suggests it may generalize well to different task types.

**Weaknesses:**

**Limited Novelty:** The paper is introducing an input-dependent update mechanism (already introduced by S6)  stabilized through the reparameterization framework and key equations (Eqs. 6, 7, and 8) borrowed directly from StableSSM [1], raising concerns about the originality of the contributions. The paper only states that it was “inspired” by stable reparameterization, yet much of the core methodology relies on prior work.


**Inconsistent Notation:** The notation for $𝐴_k$ is unclear, with dependency on input appearing inconsistently (e.g., it appears in Eq. 5 and line 267 -- e.g. $𝐴_k(u_k, \theta_m)$ but is omitted elsewhere -- e.g. $𝐴_k(\theta_m)$. This lack of uniformity in notation undermines the model’s theoretical presentation.

**Weak Justification for S5 Model Selection:** S5 is mentioned as the basis for S7, but no rationale is provided for not using S6 (Mamba) and the reparameterization technique from StableSSM. Moreover, no connection or description of the S5 model is given (MIMO approach etc.)

**Assumptions Clarity:** Assumptions (3.1, 3.2, and 3.3) are not well justified or examined for feasibility, and the text lacks guidance on implementing or verifying these assumptions. This leaves important theoretical aspects of the model unaddressed.

**Unclear Contribution of Neuromorphic Design:** The neuromorphic-specific design choices in Section 3.4 seem disconnected from the rest of the model’s development (no other mention on the first part of the paper). It’s unclear whether these additions (Eq. 11, 12) apply exclusively to neuromorphic tasks or extend to other benchmark tasks.

**Lack of Benchmark Justification:** The paper does not clarify why specific datasets were chosen. For instance, given the input-dependent nature of S7, benchmarks used by similar models like Mamba (e.g., Selective Copy or Induction Heads or other similar benchmarks -- see Section 7/Table 4 of the thematic survey [2]) might have been more appropriate for comparison.

**Poor Performance on LRA Benchmarks:** S7’s subpar performance on LRA benchmarks raises concerns about its applicability to heterogeneous data. The authors provide only a brief discussion, without substantial insight or proposed solutions for improving performance on these challenging tasks.


[1] Wang, Shida, and Qianxiao Li. "Stablessm: Alleviating the curse of memory in state-space models through stable reparameterization." arXiv preprint arXiv:2311.14495 (2023).

[2] Tiezzi, Matteo, et al. "State-Space Modeling in Long Sequence Processing: A Survey on Recurrence in the Transformer Era." arXiv preprint arXiv:2406.09062 (2024).

**Questions:**

1. **Could the authors clarify the novelty of the reparameterization?** How does it differ from StableSSM’s reparameterization framework?

2. **Why was the S5 model chosen as the basis for S7?** Given that S6 with the StableSSM reparameterization might provide similar benefits, what informed this design choice?

3. **Could the authors specify if the neuromorphic-specific design applies solely to neuromorphic tasks or to all benchmarks?** This would improve clarity regarding the model's consistency across different tasks.

4. **What is the author’s perspective on improving the model's performance on data whre time-dependance is note relevant?** Given S7’s limited success on LRA, is there a feasible modification that could address these challenges while preserving input-dependence?

---

### Official Review · Reviewer_jZ6a · 2024-11-04

**Soundness:** 2
**Presentation:** 2
**Contribution:** 1
**Rating:** 3
**Confidence:** 4

**Summary:**

This paper proposes an SSM (state space model) architecture call S7, to provide an input-dependent mechanism for an existing work (S5), and showing this architecture can efficiently and accurately deal with sequence modeling tasks. The experiments show it performs better than Mamba(S6) in LRA tasks, while worse than other SSM-like models, and show it achieves good performance in neuromorphic datasets and dynamic prediction tasks.

**Strengths:**

The paper proposes an input-independent mechanism for SSM, with stable reparameterization techniques, and it provides the stability of this reparameterization through comprehensive theoretical derivations.

**Weaknesses:**

1.	The novelty is not very clear to me, as mentioned in the paper, it is claimed that this paper provides efficient input-dependent state transition dynamics based on S5, but conceptually, it is the same as what Mamba (S6) did for S4, that introduced learnable matrices A, B, C
2.	The paper claims this S7 is more efficient than Mamba, but I did not find any experiments data on the efficiency comparison with Mamba/Mamba2. Theoretically, without using parallel technologies like selective scan or others, how could one run the S7 in an efficient way when at each time step one needs to update the dynamics of A, B, C, D?
3.	The experiments do not well support the claims: the performance of S7 in LRA is substantially worse than many other methods, and there’s no results showing it’s better than mamba in general language modeling tasks (which is an important selling point of mamba-like models). This leads to a question: what is the use case of S7?

**Questions:**

1.	What does S7 mean? (is the ”7” with some particular meaning?)
Other questions please see the above weaknesses.

---

### Official Review · Reviewer_QET1 · 2024-11-04

**Soundness:** 3
**Presentation:** 2
**Contribution:** 2
**Rating:** 5
**Confidence:** 4

**Summary:**

This paper proposed a new SSM called S7 for sequence modeling. It combines the strengths of S5 (simpler structure) and S7 (input-dependent state transitions) and incorporates stable reparameterization of the state matrix. Many experiments were carried out to verify the efficiency and performance on different datasets.

**Strengths:**

1. It turns out that the combination of S5 and S6 is helpful for SSM structure and contributes to the development of SSMs.
2. Experiments are conducted on many datasets together with extensive analysis.

**Weaknesses:**

1. The core contribution of this paper seems to be successfully combining S5 and S6, with many experiments being carried out. However, the paper lacks a detailed introduction to the S5 and S6, so readers may not be completely clear about the advantages and disadvantages of these two models, and how S7 surpasses the two. E.g., why to say "S6 introduces hardware-specific complexity" (Line 88)? Some details of S5 and S6 can be shown clearly in 3.1 Background.
2. An intuitive comparison of the S4 (S4D or DSS), S5, S6, and S7 schematic can be given, to clearly show the development and difference of SSMs. Or a similar part like 4. RELATIONSHIP BETWEEN S4 AND S5 in S5 paper [1].
3. The effectiveness of Stable Reparameterization of S7 seems not to be verified.  More introduction to the initialization of the state matrix should be given, or the writing logic of Stable Reparameterization for Long-Term Dependencies (Line 211) is too hard for readers to follow.

[1] Smith, J. T., Warrington, A., & Linderman, S. W. (2022). Simplified State Space Layers for Sequence Modeling. ArXiv. https://arxiv.org/abs/2208.04933

**Questions:**

1. The S7 should combine the advantages of the S5 and S6, so it needs to be compared with both. E.g., why is it simpler than the S6? Training faster? Fewer parameters?
2. Why did the introduction of selective matrix degrade the effectiveness so much on long sequence tasks like path-X in LRA, compared to S5 (Line 402)? More analysis is needed, otherwise the S7's improved sequential modeling capabilities using input-dependent matrix don't seem useful and convincing.
3. Prior work showed that the performance of deep state space models are sensitive to the initialization of the state matrix. [1] Have you done experiments with different initializations to verify the robustness of the S7 module? Along as the experiments of effectiveness of Stable Reparameterization for Long-Term Dependencies.

[1] Albert Gu, Isys Johnson, Karan Goel, Khaled Saab, Tri Dao, Atri Rudra, and Christopher Ré. Combining recurrent, convolutional, and continuous-time models with linear state space layers. Advances in Neural Information Processing Systems, 34, 2021b.

---

### Note · Authors · 2024-11-14

**Comment:**

We decided to withdraw the paper and go for the resubmission, where we will also include language modeling experiments.

We want to thank the reviewers for the feedback, which will be considered in the future.

**Withdrawal Confirmation:**

I have read and agree with the venue's withdrawal policy on behalf of myself and my co-authors.